# Large Neutral Amino Acid Therapy Increases Tyrosine Levels in Adult Patients with Phenylketonuria: A Long-Term Study

**DOI:** 10.3390/nu11102541

**Published:** 2019-10-21

**Authors:** Alessandro P. Burlina, Chiara Cazzorla, Pamela Massa, Giulia Polo, Christian Loro, Daniela Gueraldi, Alberto B. Burlina

**Affiliations:** 1Neurology Unit, San Bassiano Hospital, 36061 Bassano del Grappa, Italy; alessandro.burlina@aulss7.veneto.it; 2Division of Inherited Metabolic Diseases, Department of Woman’s and Child’s Health, University Hospital, 35128 Padova, Italy; chiara.cazzorla@aopd.veneto.it (C.C.); pamela.massa@aopd.veneto.it (P.M.); giulia.polo@aopd.veneto.it (G.P.); christian.loro@aopd.veneto.it (C.L.); daniela.gueraldi@aopd.veneto.it (D.G.)

**Keywords:** phenylketonuria, large neutral amino acids, phenylalanine, tyrosine, treatment, blood–brain barrier

## Abstract

The standard treatment for phenylketonuria (PKU) is a lifelong low-phenylalanine (Phe) diet, supplemented with Phe-free protein substitutes; however, adult patients often show poor adherence to therapy. Alternative treatment options include the use of large neutral amino acids (LNAA). The aim of this study was to determine the Phe, tyrosine (Tyr), and Phe/Tyr ratio in a cohort of sub-optimally controlled adult patients with classical PKU treated with a new LNAA formulation. Twelve patients received a Phe-restricted diet plus a slow-release LNAA product taken three times per day, at a dose of 1 g/kg body weight (mean 0.8 ± 0.24 g/kg/day), over a 12-month period. The product is in a microgranulated formulation, which incorporates all amino acids and uses sodium alginate as a hydrophilic carrier to prolong its release. This LNAA formulation provides up to 80% of the total protein requirement, with the rest of the protein supplied by natural food. Patients had fortnightly measurements of Phe and Tyr levels over a 12-month period after the introduction of LNAA. All patients completed the 12-month treatment period. Overall, adherence to the new LNAA tablets was very good compared with a previous amino acid mixture, for which taste was a major complaint by patients. Phe levels remained unchanged (*p* = 0.0522), and Tyr levels increased (*p* = 0.0195). Consequently, the Phe/Tyr ratio decreased significantly (*p* < 0.05) in the majority of patients treated. In conclusion, LNAA treatment increases Tyr levels in sub-optimally controlled adult PKU patients, while offering the potential to improve their adherence to treatment.

## 1. Introduction 

Phenylketonuria (PKU, OMIM 261600) is an autosomal recessive genetic disorder caused by mutations of the gene encoding the phenylalanine hydroxylase (PAH, EC1.14.16.1) enzyme [1]. PKU patients are detected by neonatal screening and start a low-phenylalanine (Phe) diet soon after birth [2]. The mainstay of treatment is a lifelong low-Phe diet. The aim of dietary treatment is to prevent excessive Phe accumulation in the blood by strict control of natural protein intake in combination with the administration of a Phe-free/low-Phe protein substitute, usually based on Phe-free l-amino acids [3,4]. The PKU diet allows normal development and Intelligence Quotient (IQ), but blood Phe levels must not exceed the safe age-related range (120–360 µmol/L for 1–12 years; 120–600 µmol/L for >13 years) [2]. However, maintaining these Phe levels is often difficult for adults with PKU [5], as a Phe-restricted diet can be perceived as a great burden and a limitation on adult patients’ daily life [6]. Despite this, maintenance of good metabolic control is correlated with improved long-term cognitive and behavioral outcomes [7]. Variability of Phe concentration over time due to difficulty in adherence to the prescribed diet may also be a significant negative predictor of long-term outcomes [8].

In our recent findings, which are representative of the whole Italian PKU population, habitual consumption of low-protein foods was reported by 87% of patients, while the mean use of amino acid mixtures was low: only 18.5% of patients reported an optimal consumption [9]. The most relevant factors which interfered with adherence to the consumption of amino acid mixtures were the impact of amino acid mixture use in the social context, frequency (4–5 times/day), and its palatability [9,10,11].

In recent years, alternative or complementary treatments were proposed. New protein substitutes, including large neutral amino acids (LNAA) and glycomacropeptide, and pharmacological treatments, such as sapropterin dihydrochloride (BH4) and phenylalanine ammonia lyase enzyme, were successful in the treatment of adult PKU patients [12,13,14].

In the pathophysiology of PKU, modification of amino acid transport across the blood–brain barrier (BBB) plays an important role with the increase of brain Phe influx competing with other LNAAs at the same transport level [15]. Therefore, LNAA supplementation may have several important functions: reducing brain Phe concentrations, increasing brain neurotransmitter concentrations, and increasing brain essential amino acid concentrations [16]. LNAA are transported across the blood–brain barrier (BBB) by a family of transport proteins called L-system, which contain LAT1, a catalytic subunit, and a type II glycoprotein subunit called 4F2hc. LAT1-4F2hc is selective for LNAA and is essential for the BBB transport [17]. The role of LNAA and their transport into the brain in PKU patients was reported in 1953 by Christensen, who proposed that high blood concentrations of Phe could interfere with the transport of other LNAAs into the brain [18].

The LNAA transporter facilitates the transport of LNAA across the BBB. Elevated blood levels of Phe is predicted to increase the uptake of Phe at the expense of other LNAAs, resulting in the disruption of brain amino acid homeostasis. In PKU patients, administering large amounts of Phe-free LNAAs competes with Phe and may reduce the influx of Phe into the brain [19].

Nowadays, two different LNAA treatment strategies are reported in the literature: LNAA alone or in combination with a low-Phe diet [16]. All studies that evaluated the use of LNAA in PKU patients were mainly focused on monitoring Phe plasma levels during a very limited time period (less than 6 months) and included mixed cohorts of patients, mostly children and adolescents. In this age range, diet is under the control and/or influence of parents. Nevertheless, a significant reduction of plasma Phe for a short period of time was reported in two studies [20,21], but other authors did not report this effect [19,22,23,24].

The aim of our study was to evaluate plasma Phe, Tyr, and their ratio in a cohort of sub-optimally controlled classical adult PKU patients treated with a new LNAA formulation combined with a low-Phe diet.

## 2. Materials and Methods

### 2.1. Study Population

In total, 12 adult PKU patients from the Division of Inherited Metabolic Diseases, University Hospital of Padova, Italy, were selected to participate in the study because of their persistently low diet adherence due to their refusal to take supplementation with an amino acid mixture. These patients were already on a Phe-restricted diet and amino acid supplementation covering 80% of their total protein intake.

Diagnosis of PKU was performed by neonatal screening, and all patients were treated with a Phe-restricted diet from birth. All 12 patients had a PAH genotype and BH4 loading test showing a lack of response to the PAH cofactor sapropterin dihydrochloride.

Dietary intake was not recorded during the study due to its long treatment period (i.e., 12 months); therefore, detailed information on dietary habits is not available.

The principles of Good Clinical Practice were adhered to throughout the study, in accordance with the Declaration of Helsinki and ICH/GCP. The study was approved by the Ethical Committee of Padua University Hospital, Italy. Written informed consent was obtained from all participants.

### 2.2. LNAA Supplementation

The LNAA formulation used in this study was NeutrAfenil^®^ Micro R (PIAM). This product is registered in Italy for the treatment of PKU under medical prescription, and it is under registration in Turkey and Spain. Because the product does not contain vitamins or minerals, all patients in our study were regularly supplemented with multivitamins and minerals.

The LNAA formulation is a granulation that can be easily swallowed with water or other liquids. The amino acids are incorporated in microgranules that are coated with a methylcellulose film to prevent any unpleasant taste related to the amino acid mixture. The cap of the bottle can be used to measure 17 grams of amino acids.

The most significant characteristics of the LNAA formulation are improved palatability (unflavored taste), its prolonged-release formulation due to sodium alginate as a hydrophilic carrier that provides a physiological absorption similar to that of natural protein, and the high concentration of Tyr in comparison with other available products.

The composition of the LNAA formulation is shown in Table 1

### 2.3. Study Design

The demographic characteristics of the patient population are shown in Table 2. Patient age ranged from 19 to 38 years (mean ± standard deviation (SD) was 29.6 ± 6.8 years), most patients were educated to a high school or University level, and three-quarters of the patients were employed.

The LNAA supplement was taken three times per day over a 12-month period (breakfast, lunch, and dinner), at a dose of 1 g/kg body weight. No other amino acid mixture was given to the patients and no changes were made to their low-Phe diet.

Patients collected fasting blood spots at home every two weeks in the morning before taking LNAAs. Measurements of Phe and Tyr levels were done on these dried blood spots (DBS) using liquid chromatography/tandem mass spectrometry, and the Phe/Tyr ratio was calculated. The consumption of LNAA was calculated by the dietitian every six months. All patients stated that, due to their commitments, they generally had lunch or dinner out of the home, with a high intake of Phe (3–4 times the prescribed value).

### 2.4. Statistical Analysis

The Wilcoxon paired sample T-test was adopted to compare data sets acquired from the patients before and after the start of LNAA therapy. A *p*-value of <0.05 was considered significant. Categorical variables are expressed as *N* (%), and quantitative variables are expressed as mean ± SD.

## 3. Results

Patient nutritional data prior to and after LNAA were introduced, including the total protein intake, medical foods intake, and Phe and Tyr intake, are summarized in Table 3. In general, the nutrient profile of LNAA treatment was constant during the 12-month period, whereas the amino acid mixtures–medical foods (AAM–MF) intake was often refused by the patients. The total Tyr intake (g/day and mg/kg) in LNAA was significantly higher than in the standard AAM–MF (*p* = 0.008 and *p* = 0.0039, respectively); no significant differences were identified for the other components.

In the 12-month period prior to the introduction of LNAA, the mean ± SD Phe and Tyr values ranged from 628 ± 148 to 1033 ± 198 μmol/L and from 32 ± 7 to 87 ± 35 μmol/L, respectively, while the mean ± SD Phe/Tyr ratio ranged from 9.7 ± 2.2 to 19.9 ± 2.8 μmol/L, and the DBS frequency ranged from 5 to 38 (median = 19): 3 patients (25%) sent more than 24 DBS samples. Phe and Tyr values, Phe/Tyr ratio, and DBS frequency are shown for each patient in Table 4 and are summarized for the overall population in Table 5. The molecular analysis of each patient is shown in Table 4.

All 12 patients completed the 12-month trial. At baseline and at the end of the study, patients had a mean ± SD weight of 70.0 ± 18 kg and 70.4 ± 18 kg, respectively, and a mean ± SD BMI index of 24.80 ± 4.83 and 24.92 ± 5.11, respectively. Neither weight nor BMI were significantly different when compared prior to and after the introduction of LNAA (*p* = 0.57 and *p* = 0.95, respectively).

Adherence to LNAA supplementation was very good.

Table 4 and Table 5 also show the mean values of Phe, Tyr, and mean Phe/Tyr ratio and frequency of DBS measurements for each patient and for the overall population, respectively, over the 12-month treatment period. Overall, the mean ± SD Phe and Tyr values ranged from 736 ± 93 to 1269 ± 265 μmol/L and from 59 ± 15 to 108 ± 35 μmol/L, respectively, while the mean ± SD Phe/Tyr ratio ranged from 8 ± 1.4 to 20.6 ± 7.6 μmol/L. The frequency of DBS ranged from 14 to 39 (median = 23.5) over the 12-month treatment period: 7 patients (58%) sent more than 24 DBS samples. The statistical difference in the DBS frequency pre- and post-LNAA introduction was significant (*p* = 0.0088) (Table 5).

For most patients, mean Phe levels were similar before and after the 12-month treatment period with LNAA (Figure 1), whereas Tyr levels increased significantly in 11 out of 12 patients (92%) (mean 75 μmol/L ± 16 µmol/L; *p* = 0.0195) (Figure 2). Compared with before the introduction of LNAA, the mean Phe/Tyr ratio decreased significantly in 10 out of 12 patients (83%) (mean 12 µmol/L ± 3 µmol/L; *p* < 0.05) after 12-months of treatment with LNAA; in 2 patients, the Phe/Tyr ratio showed a small increase (mean 17 µmol/L ± 5 µmol/L; *p* < 0.16) (Figure 3).

We also report data which illustrates the level of Phe and Tyr at baseline, after 6 months (short term), and after 12 months (long term) of LNAA treatment for each patient (Figure 4 and Figure 5, respectively) and for the overall population (Table 6).

Compared with baseline, Phe values decreased significantly in the first six months (short-term period) (*p* = 0.00458), but then increased significantly (*p* = 0.0522) in the last six-month treatment period with LNAA (Figure 4 and Table 5). Although dietary intake was not recorded during the study, it is likely that a stricter diet regimen over the initial six-month treatment period resulted in the decrease in Phe levels at six months, whereas patients may have relaxed their dietary restrictions, causing Phe values to increase at 12 months. On the other hand, Tyr levels tended to be stable over time, with no significant increase over the short-term period (baseline to six months; *p* = 0.324) or from 6 to 12 months (*p* = 0.1400) (Figure 5), suggesting that consumption of the new LNAA formulation was consistent.

## 4. Discussion

In this study, blood Tyr levels were significantly elevated in the patient cohort for the entire duration of the study. The mean percentage variation increased after 12 months of treatment, and the difference in Tyr levels compared with baseline was statistically significant. Even in the short-term, Tyr values showed an increase, which was also confirmed in the long-term (12 month), with an increase in 67% of patients. This increase also reflects the high amount of Tyr present in the LNAA formulation in comparison with other available products. Changes of Tyr levels were also confirmed by the decrease in Phe/Tyr ratio during all periods of the study.

In our study, mean blood Phe levels did not change significantly in the 12 months before starting LNAA therapy compared with the 12-month period after the initiation of LNAA therapy. Indeed, patients still showed poor control over Phe levels. These data support previous studies [24] that did not show any change in blood Phe levels after LNAA administration. In a previous study conducted in a different cohort of PKU patients, we observed a reduction of plasma Phe concentration after one week of treatment with a different LNAA product [20].

In our study, we confirmed these results over a short-term period, but a decrease of Phe values was not confirmed over a long-term period (12 months). Indeed, at the end of this study, Phe levels returned to high levels in 75% of patients. 

Blood Phe concentration remained unchanged after one and six months of LNAA supplementation in a small cohort of six patients with classical PKU [23]. Nonetheless, an increase in blood Tyr and tryptophan (Trp) levels was observed, while proton magnetic resonance spectroscopy detected a decrease in brain Phe levels. Despite the correlation between the use of LNAA and brain Phe concentrations, blood Phe levels do not appear to be influenced by ingestion of the LNAA mixture [23].

In a previous study, which focused on adult Italian patients, we showed that dietary adherence was poor, with reduced daily use of amino acid supplements (<4–5 times/day in 82% patients) [9]. The survey also showed that the mean daily use of amino acid mixtures was low due to patient embarrassment in consuming mixtures when out of home or while traveling, the palatability of the mixture, and the number of administrations required.

The use of LNAAs in the treatment of patients with PKU was suggested as early as 1948 and offers an alternative complementary treatment option; however, the optimal composition of LNAA is still unknown [25].

Studies that have evaluated LNAA in PKU patients [26,27] are relevant; however, they failed to show any clear advantage in terms of a reduction in brain Phe concentrations. A four-week study performed with 16 classical PKU patients on a strict daily food diary identified a trend toward lower plasma Phe levels with LNAA supplementation, suggesting that LNAA supplementation may be beneficial in patients who are unable to comply with PKU medical products [28]. The authors suggested that the reduction of plasma Phe levels with LNAA supplementation may be due to competition with Phe at the level of transport across the gut, but without direct measurement of brain/gut levels, these findings are speculative.

In 2006, Matalon and colleagues evaluated a new formulation of LNAA (with administration of 0.5 to 1 g/kg/day) and reported that blood Phe concentrations decreased after one week without any change in dietary practice [21]. A following study showed a significant reduction of Phe blood concentrations (average reduction of 39%) in all 20 patients treated with LNAA supplements compared with a decline of 5.4% in the placebo group [20].

The most recent report on the use of LNAA demonstrated a positive correlation between blood Phe reduction and LNAA intake in 12 PKU children after four weeks of treatment [22]. Notably, patients who had a smaller reduction in Phe levels were adolescents. The authors hypothesized that younger children were able to follow their diet more strictly because of their parents’ supervision, whereas parental supervision was less effective over adolescents [22].

Data on brain Phe levels in patients treated with LNAA are scarce. Pietz et al. showed that LNAA had the ability to block Phe transport into the brain [19]. In this study, they used an oral purified L-Phe dose, with or without LNAA, and measured the influx of Phe into the brain, using proton magnetic resonance spectroscopy. They showed that cerebral Phe concentrations remained unchanged or decreased during the concomitant LNAA consumption, even if the plasma Phe levels increased.

In animal experiments using PKU mice on a normal diet supplemented with LNAA, van Vliet and colleagues showed significantly reduced brain Phe concentration and increased brain serotonin and norepinephrine, but no significant changes in dopamine concentration [15]. The same authors also showed that, in PKU mice, LNAA were as equally effective as a severe Phe-restricted diet in restoring brain monoamine concentrations, thus indicating that this therapy could be a promising alternative strategy for treating adult PKU patients who are unable adhere to a severe Phe-restricted diet [29]. It is noteworthy that the accumulation of Phe even in early treated patients can lead to cerebrospinal fluid neurotransmitter deficits, with a reduction of homovanillic acid and 5-hydroxyindoloacetic acid [30,31]. Therefore, the possibility of modifying the therapeutic approach by reducing the flux of Phe into the brain, with the mechanism of LNAA competition, could offer a major benefit in the treatment of the disease.

The change in LNAA levels detected in plasma and the brain reflects the different distribution of LNAA in plasma, in the neurovascular unit, and the actions of the specific transporter systems across the BBB. The PAH, which converts Phe into Tyr, is not expressed in the brain. Experimental models showed that Tyr is the LNAA with the most consistent correlation between serum ratio and brain tissue levels [32]. Indeed, all Tyr available to the brain should be transported from the blood. On the other hand, the kinetics of Phe are unique, with the highest affinity for BBB transport in the rat brain [33].

The complexity of the neurovascular unit, which comprises the BBB endothelium, the astrocytes, and neurons, also makes any direct comparison between LNAA data achieved from experimental models with those of the human brain difficult [34].

Most patients in our cohort showed nonsignificant biochemical changes (blood Phe levels) at the end of the LNAA treatment period, which is contradictory to some of the previously published studies. However, previous studies were carried out over very short time periods (less than 12 months), whereas patients in the current study were treated with LNAA for a 12-month period. Indeed, in the short term (less than 6 months), a decrease in Phe levels was detected, but Phe levels returned to baseline levels for the majority of patients by the end of the 12-month treatment period (Figure 4).

Another possible contributing factor to the nonsignificant reduction of Phe levels is the age of the patients. In our cohort, the patients were all adults, whereas, in the Concolino study, they were pediatric patients, with a better therapy adherence [22].

Limited data are available regarding Tyr values and Phe/Tyr ratio in patients on LNAA supplementation. The LNAA product used in the current study is very rich in Tyr and Trp in comparison with other products on the market and offers a new extended-release formulation. Both these characteristics can help avoid fluctuation of blood Phe in adults. Phe fluctuations were already suggested as a major source of neurotransmitter perturbation, with subsequent behavioral complications for PKU patients [35]. More recently, an in silico model, simulating the PKU condition, suggested that LNAA supplementation could contribute to maintaining Phe at normal physiologic levels, avoiding inappropriate Phe distribution inside the neurovascular unit [36].

Moreover, high levels of Tyr may promote an increased brain monoaminergic neurotransmitter concentration [37]. Three mechanisms driving the improvement observed with LNAA administration were confirmed in a murine PKU model: LNAA supplementation reduced brain Phe levels, increased brain levels of non-Phe LNAA, and increased neurotransmitter levels [15].

Results from our study suggest that the period of biochemical and clinical follow-up should be increased in comparison with what was known from previous studies. It is possible that a follow-up of 12 months is still not long enough for a chronic condition like PKU. At present, we are continuing to monitor the patients receiving LNAA treatment.

## 5. Limitations

There are limitations within the current study. Firstly, the results of our study are related to a relatively small group of noncompliant adult PKU patients over a long period of time. It is difficult to compare our observational study with other published studies in patients (children and adults combined) conducted over a short period of time. Although some findings may be a consequence of bias and lack of an optimal composition of LNAA, a follow-up study with an increased number of noncompliant patients should be performed before concrete conclusions can be drawn.

Secondly, food intake and compliance were not regularly recorded and obtained through self-reporting. The authors believe that this closely reflects the real life of adult PKU patients; however, patient-centered data collection presents opportunities for misreporting. The ability to conduct a research study is particularly difficult when the patients are predominantly noncompliant PKU and are disengaged from active metabolic follow-up, highlighting the challenges in their inclusion in relatively complicated research procedures. This has done recruitment particularly difficult and the design of a randomized control trial study complex, as recently reported by Green et al. [38].

Thirdly, the plasma Phe, Tyr, and their relative ratio are clinically effective biomarkers for the follow up of the disease, but due to the lack of standardization of blood sampling obtained through self-completed blood samples, caution should be taken when extrapolating the results of this study.

## 6. Conclusions

In conclusion, LNAA supplementation to a low-protein diet should be considered in adult PKU patients who refuse amino acid mixtures. The use of a different therapeutic option, such as LNAA with its tasteless formulation, showed two major results: the increase of blood Tyr levels and the improvement of patient adherence to medication. Further studies are needed to assess the nutritional intake and metabolic status of PKU patients using LNAA for longer periods of time (i.e., 24–36 months).

## Figures and Tables

**Figure 1 nutrients-11-02541-f001:**
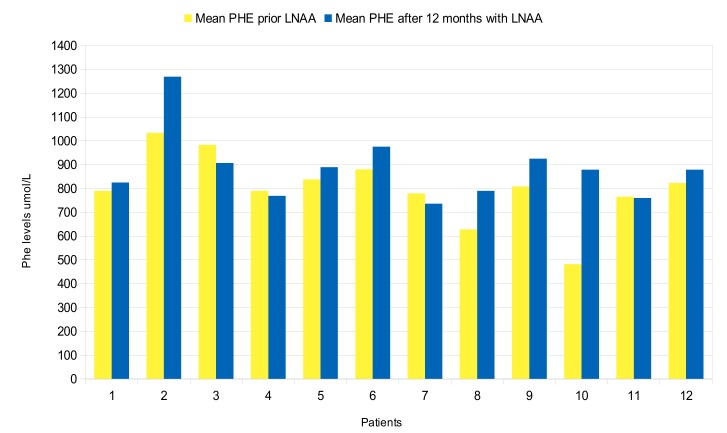
Mean Phe levels 12 months prior and after LNAA introduction for each patient.

**Figure 2 nutrients-11-02541-f002:**
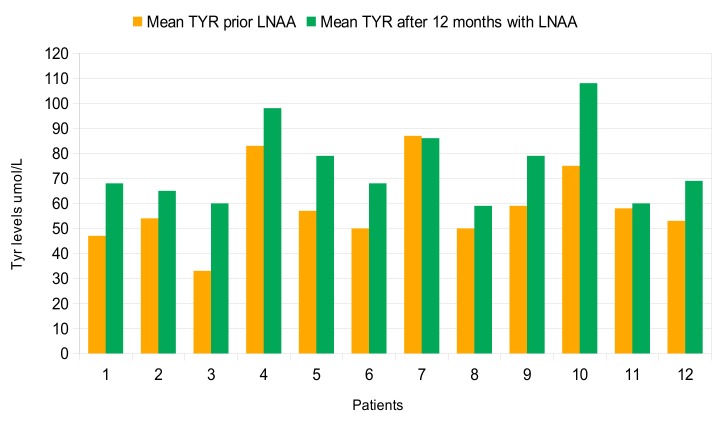
Mean Tyr levels 12 months prior and after LNAA introduction for each patient.

**Figure 3 nutrients-11-02541-f003:**
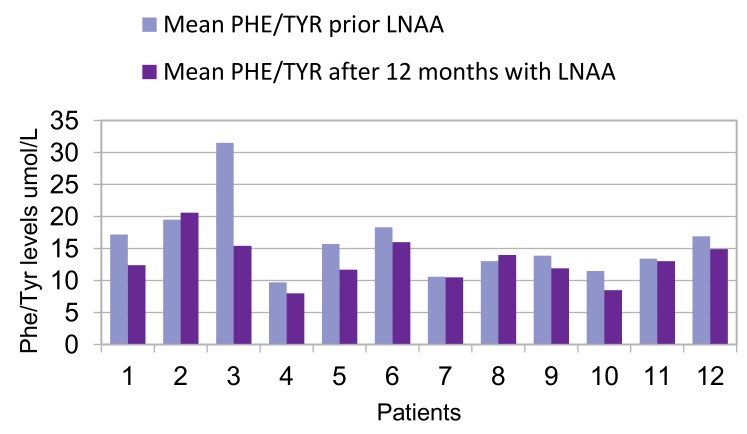
Mean Phe/Tyr ratio 12 months prior and after LNAA introduction for each patient.

**Figure 4 nutrients-11-02541-f004:**
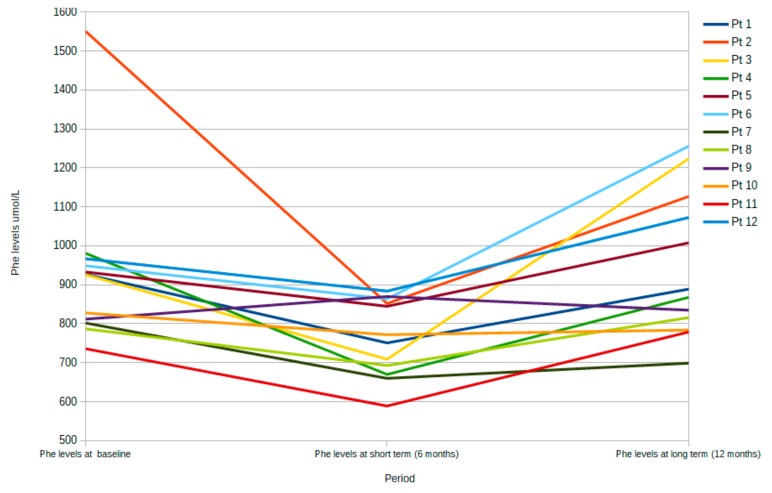
Phe level at each time point: baseline, after 6 months (short term) and 12 months (long term) of LNAA treatment for each patient.

**Figure 5 nutrients-11-02541-f005:**
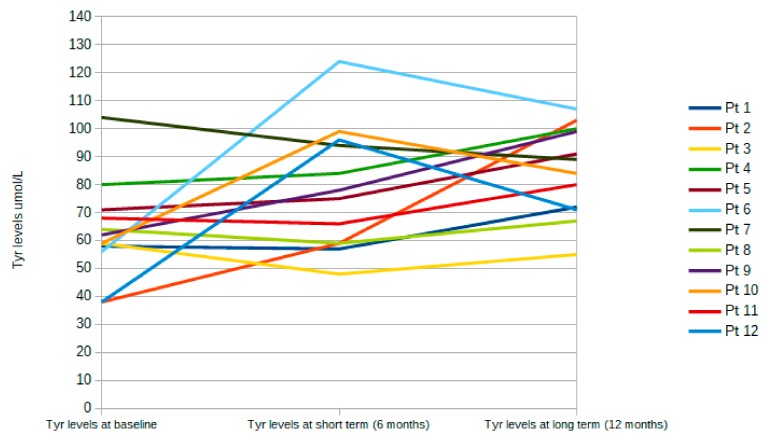
Tyr level at each time point: baseline, after 6 months (short term) and 12 months (long term) of LNAA treatment for each patient.

**Table 1 nutrients-11-02541-t001:** Large neutral amino acids (LNAA): nutritional information per 100 g.

Energy	1686 kJ
	399 kcal
Fat	5.3 g
of which saturated fat	5.3 g
Carbohydrates	12 g
of which sugars	0 g
Fiber	5.8 g
Equivalent protein	70.79 g
Salt	1.6 g
L-Arginine	1.92
Aspartate	4.95
L-phenylalanine	0 g
L-isoleucine	10 g
L-histidine	3.36 g
L-leucine	12 g
L-lysine	5.44 g
L-methionine	2.72 g
L-tyrosine	24 g
L-threonine	2.56 g
L-tryptophan	8 g
L-valine	10 g

**Table 2 nutrients-11-02541-t002:** Demographic characteristics of all patients.

		*N* (%)
Gender	Female	5 (42)
	Male	7 (58)
Age range	19–38	
Marital status	Married/with partner	6 (50)
	Unmarried	6 (50)
Education level	Middle	1 (8)
	High	7 (58)
	University	4 (33)
Employment status	Employed	9 (75)
	Unemployed	1 (8)
	Student	2 (17)
Leisure time	Sport	3 (25)
	Social activity	2 (17)
	Art, museum, exhibitions	2 (17)
	Not specified	5 (42)

Abbreviations: LNAA, large neutral amino acids; *N*, number.

**Table 3 nutrients-11-02541-t003:** Composition of standard amino acid mixtures compared with LNAA.

	AAM–MF		LNAA–MF	
	Mean ± SD	Median	Mean ± SD	Median
Total proteins (g/day)	72 ± 11	70	63 ± 13.45	59
Total proteins (g/kg/day)	1 ± 0.23	1.13	0.8 ± 0.24	0.75
MF (g/day)	51 ± 7.36	52	42 ± 9.78	38
MF (g/kg/day)	0.7 ± 0.23	0.8	0.5 ± 0.21	0.4
Phe intake (mg/day)	834 ± 455.41	700	709 ± 199.91	700
Phe intake (mg/kg)	12 ± 4.81	10.3	10 ± 1.8	10.2
Tyr intake (g/day)	4.8 ± 0.72	4.6	6.6 ± 0.61	6.5
Tyr intake (mg/kg)	73 ± 21.54	74.7	100 ± 26.1	103.7

Abbreviations: AAM, amino acid mixtures; LNAA, large neutral amino acids; MF, medical foods.

**Table 4 nutrients-11-02541-t004:** Molecular analysis of each patient and the mean (SD) values of Phe, Tyr, Phe/Tyr ratio, and DBS frequency over the 12-month period prior to the introduction of LNAA and over the 12-month LNAA treatment period.

Patient	Molecular Analysis	12-month Period Prior to LNAA				12-month LNAA Treatment Period			
		Mean Phe Values ^a^ (SD)	Mean Tyr Values ^a^ (SD)	Mean Phe/Tyr Values ^a^ (SD)	DBS Frequency	Mean Phe Values ^a^ (SD)	Mean Tyr Values ^a^ (SD)	Mean Phe/Tyr Values ^a^ (SD)	DBS Frequency
1	c.473G > A/c.1315 + 1G > A	790 (80)	47 (8)	17.2 (2.5)	8	825 (114)	68 (10)	12.4 (2.4)	17
2	c.473G > A/c.526C > T	1033 (198)	54 (10)	19.5 (4.4)	5	1269 (265)	65 (14)	20.6 (7.6)	14
3	c.473G > A/c.473G > A	983 (142)	32 (7)	31.5 (7.3)	10	907 (166)	60 (8)	15.4 (3)	20
4	c.842 + 3G > C in heterozygosis	790 (118)	83 (14)	9.7 (2.2)	18	769 (144)	98 (18)	8 (1.4)	34
5	c.47_48delCT/c.1315 + 2T > C	838 (179)	57 (12)	15.7 (6.6)	20	889 (149)	79 (19)	11.7 (2.7)	24
6	c.842C > T/c.1315 + 1G > A	880 (160)	49 (11)	18.3 (4)	38	975 (148)	68 (29)	16 (5)	39
7	c.842C > T/c.1315 + 1G > A	779 (108)	87 (35)	10.6 (5)	25	736 (93)	86 (41)	10.5 (4.9)	27
8	c.842C > T/c.1315 + 1G > A	628 (148)	50 (13)	13 (5.2)	26	790 (147)	59 (15)	14 (4.5)	29
9	c.1222C > T/c.1315 + 1G > A	808 (135)	59 (7)	19.9 (2.8)	12	925 (139)	79 (13)	11.9 (1.8)	18
10	c.782G > A/c.782G > A	842 (94)	75 (12)	11.5 (2.3)	23	879 (91)	108 (35)	8.5 (1.9)	36
11	c.782G > A/c.1066 − 11G > A	765 (234)	58 (11)	13.4 (3.9)	23	760 (124)	60 (11)	13 (2.7)	24
12	c.1222C > T/macro deletion in exon 3	823 (117)	53 (19)	16.9 (5)	18	1000 (163)	68 (17)	15.2 (3.7)	15

Abbreviations: DBS, dried blood spots; Phe, phenylalanine; SD, standard deviation; Tyr, tyrosine. ^a^ Expressed in μmol/L.

**Table 5 nutrients-11-02541-t005:** Phe, Tyr, Phe/Tyr ratio, and DBS frequency over the 12-month period prior to the introduction of LNAA and over the 12-month LNAA treatment period for the overall patient population.

Overall Patient Population	12-Month Period Prior to LNAA	12-Month LNAA Treatment Period	*p*-Value
Phe μmol/L, mean (SD)	752 (143)	894 (145)	0.0522
Tyr μmol/L, mean (SD)	59 (13)	75 (16)	0.0195
Phe/Tyr ratio μmol/L, mean (SD)	16 (4)	13 (3)	0.049
DBS frequency	19 (9)	25 (8)	0.0088

**Table 6 nutrients-11-02541-t006:** Phe and Tyr values at baseline, 6 months and 12 months after the LNAA treatment period for the overall patient population.

Overall Patient Population	12-Month Period Prior to LNAA	6-Month LNAA Treatment Period	12-Month LNAA Treatment Period
Phe μmol/L, mean (SD)	752 (143)	645 (101)*	894 (145) **
Tyr μmol/L, mean (SD)	59 (13)	65 (16)*	75 (16) **

* *p*-value: Phe *p* = 0.00458, Tyr *p* = 0.324. ** *p*-value: Phe *p* = 0.00374, Tyr *p* = 0.1400.

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
