# Peer review of "Large Neutral Amino Acid Therapy Increases Tyrosine Levels in Adult Patients with Phenylketonuria: A Long-Term Study"

_nutrients, 2019, doi:10.3390/nu11102541_

Round 1

Reviewer 1 Report

General comment: First, I would like to congratulate the authors for this important piece of work. Not much data is available on patients taking LNAA, so all data is important. I believe that some changes would improve this manuscript. 

Title: I would like to suggest that the title specifically states that this is a new formulation/or slow release of LNAA as it could be mistaken with the normal LNAA products used. I would also prefer to have something more general like "metabolic control of PKU patients" rather than only focusing in the tyrosine results.

Authors list: Not sure if Cristian Loro is a mistake as the email states christian.loro@aopd.veneto.it

Abstract

Some corrections:

Line 14: however adult patients (without coma)...

Line 16: Phe/tyrwihout The ... adult patients with Classical PKU

Line 20: erase "and the Phe/Tyr ratio...

Line 24: should read: in all patients. Consequently, Phe/Tyr....

Line 26: medication should be amino acid mixture or protein substitute.

General comment: Could perhaps add mean levels and p=values...

Introduction

Some corrections:

Line 36: Phe-free/ low-Phe protein substitute

Line 37: The PKU diet instead of this diet. Please describe definition of IQ first time it appears.

Line 49: erase the difficulty of the frequent administration to frequency

Line 50: New protein substitutes (instead of special dietetic products)

Line 66-67: This sentence is difficult to understand. Could you please clarify what you mean by LNAA supplementation to a low Phe diet and LNAA supplementation combined with conventional PKU diet therapy. I think it is not clear as it is...

General comments: I think it should be added to the introduction more information on the different composition of LNAA compared with standard AA used and that the optimal composition of LNAA is not known. There is little data on LNAA although it is used by a lot of adult patients when they have poor adherence to the PKU diet.

Material and methods

Some corrections:

Line 78: recruited - Do you mean screened?

Line 82: Adherence to

line 86: erase "due to their refusal to take a supplementation with an amino acid mixture"

line 89: adhered throughout - erase to

LNAA supplementation

line 99: At this level, the polymer...

Line 102: Last sentence should come at the beginning of this subheading...

General comment: maybe another subheading with AA used by patients as part of the baseline assessment should be added...

Study protocol - maybe should be changed to study design

Line 105: should read a 12-month period (breakfast, lunch and dinner time).

Line 108: Erase the sentence " Patients were followed for 12 months after introduction of LNAA. - It's a repetition. 

Line 109: Erase " and the Phe/Tyr ratio". This is calculated from the measurements of Phe and Tyr.  

Line 112: correct to " dinner out of home"

Line 114: erase "and clinical findings"

line 114 to 117: I think this paragraph should only come in the results. 

Table 1: I think the title could be " Demographic characteristics of all patients". I would also erase the last variable "treatment type". I don't think it adds relevant information. Also please check % match when number 2. Should round always the same way.   

General comment: I would start the subheading study protocol first with the measurements and then explain LNAA intake. So, would change the order of the paragraphs. I think maybe it would also help if a little figure was added explaining the study design. This would help the reader understand the study easily.

Statistical analysis 

Line 122: should read "T-test was..."  

General comment: Could you please state what program did you use to do the statistical analysis...

Results

Line 139:  change execution to measurements …

General comment: I think it would be good to combine table 2 and 3. I know it will be a big table but it will be easier for the reader to compare the measurements of the same patient on the different moments of the study… It would be also interesting to add DBS frequency on the first 12 months prior to LNAA introduction.

Figure 1: y axis label – change values to Phe levels. Maybe it should be better to state Mean Phe 12 months prior to LNAA on the legend. Title of the figure could be “Mean Phe levels 12 months prior and after LNAA introduction for each patient”.

Figure 2 and 3: Same comments as Figure 1.

Line 171-174: It is not clear if the levels are mean levels during 6 months or the level at each time point (baseline/6 months/12 months). Not clear in the text and on the graphs.

Figure 4: y axis label – change values to Phe levels. Maybe would also define in legend short term and long term.

Figure 5: Same comments as on figure 4. Why isn’t there any data for patient 12?

General comment: Even though I am aware you don’t have dietary intake data, it would be good if you could present tyrosine intake from AA vs tyrosine intake from LNAA. It would help discussing the tyrosine results as well.

Discussion

I think the discussion needs major changes.

I think it would be better to start stating the main results of the study and then discuss it with what was found previously in the literature. Maybe discussing one by one the main results:

- Phe levels;

-Tyr levels (these also need to be discussed comparing tyrosine levels from AA vs LNAA);

- Phe/tyr (discuss results and also the significance and importance of this ration that is not used in all clinics).

It would also be important to add a limitations part to the discussion with the following points:

- Maybe non-adherence to previous protein substitute may influence the results so it would not be comparable;

- Lack of dietary intake data which may also influence the metabolic control;

- No data on nutritional biochemical markers.

It would be important to state as well if this is the first time this product is reported in the literature?

This product has no vitamin and minerals so it would be good to emphasize the importance of measuring nutritional markers and vitamin and mineral status in the future. For future research, it would be also crucial to have detailed data on the dietary intake of these patients.

It would be also good to underline the need for more data using LNAA as very little data is available in the literature and this is a type of product that is widely used when patients start to be non-complaint with the PKU diet.

Specific corrections:

Line 184-194: It is a repetition of the introduction

Line 255: Do you think maybe compliance improved in the first 6 months and then deteriorated again by the end of 12 months? Maybe they complied with LNAA in the beginning and then compliance started to be similar to AA?

Line 260-262: Repetition of lines 253-255.

Line 292: “adherence to the PKU treatment.”

Line 293: all patients. Consequently,  

Conclusion:

Maybe could add that: It would be important to have more data on metabolic control, nutritional intake and nutritional status when PKU patients are using LNAA.

Reviewer 2 Report

This study in the scope of PKU treatment was generally well written and structured. However, in my opinion the paper lacks nutritional information that could bring insight into the results found.

I made the following suggestions and questions:

Lines13-14, Abstract: I would add “The standard treatment for phenylketonuria (PKU) is a lifelong low-phenylalanine (Phe) diet supplemented with Phe-free protein substitutes”; Lines 122-123, Statistical Analysis: Did you test the normality of the variables? Which test did you use for that? Kolmogorov-Smirnov or Shapiro-Wilk? I suggest “A paired sample t-test” instead of t-test only. I also suggest a sentence like this “Categorical variables are expressed as N (%) and quantitative variables as mean ± standard deviation (SD)”. I highly recommend completing this section; Lines 135-137, Results: Where are the p-values for weight and BMI? Lines 137, Results: How did you measure/ evaluate adherence to LNAA? Discussion: I think authors should have collected data on dietary intake during appointments. This paper illustrates a 12 month-interventional study, but the outcome measures were few taking into account the time of intervention. The nutritional data might have helped to interpret the results. In this sense, I propose putting a sentence with limitations/shortcomings of the study.

Given these points the manuscript requires minor revisions.

Reviewer 3 Report

Overall the paper adds to the growing evidence of using LNAA. It challenges the difficulty of non- adherent adolescents and adults with PKU, and suggests a possible alternative treatment modality, but one needing further long term studies and evaluation.

I have made some suggestions to improve the delivery of the paper, these are minor/moderate changes.

Further comments are in the word document attached:

Review of tables and formatting of results Additional section on the product, compositional details  and its delivery Section on limitations of the study Minor queries and re-wording of some sentences

Reviewer 4 Report

Thank you for the opportunity to review the paper of Burlina et al. The manuscript is well written in parts, but poorly structured. The paper focuses on a key and understudied population of PKU patients and the authors have done well to keep all sub-optimally controlled patients 100% compliant and on trial throughout the 12-month intervention period. I think this has potential to be clinically important work, however, I have several concerns (detailed below) that need to be addressed before the paper can be accepted. Unfortunately, there are several key limitations to the study which the authors fail to recognise and acknowledge.

Abstract

Line 13-14. The authors are correct the PKU diet is a lifelong low-phenylalanine but also includes intake of protein substitutes. Can the authors add this to the sentence? It will nicely introduce the second sentence. Line 17-18. Please add further detail about the intervention period. For example, the fact the LNAA supplement was taken three times per day over a 12-month period as this is not currently communicated in the abstract. Line 18. Can the authors provide an average protein equivalent that patients’ consumed as a result of (1 g/kg/day)? Line 22-23. The authors mention adherence was good compared to patients’ previous amino acid mixture. Please provide the statistics behind this statement and add this information for the reader to justify the authors’ conclusions. Line 23. “Taste was a major complaint by patients.” Was this for the trial product or patients’ previous amino acid mixture? Line 23-25. Please provide the data for phenylalanine, tyrosine and phenylalanine:tyrosine ratio. Line 25-26. The concluding sentence of the abstract does not really match the title of the manuscript. Would something like ‘LNAA treatment increases tyrosine levels in sub-optimally controlled adult PKU patients while offering potential to improve their adherence to medication’ be a better representation of the results/conclusions?

Introduction

Line 45-47. To give further breadth provide comment on how your recent statistics of 18.5% of patients report optimal consumption of amino acid supplements this is similar to other countries (i.e. US, UK, Australia) and provide references. Line 49. Provide reference at the end of the sentence. Line 62. The transporter that takes LNAAs into the brain is the LAT-1 carrier system. Please amend. I feel the term ‘treatment’ is inappropriate. The use of amino acid mixtures is to help manage PKU, not treat it. I recommend authors change this terminology throughout. From an intervention perspective this could be referred to as ‘…adult PKU patients after receiving a new LNAA…’ Line 74-75. Add that the cohort are sub-optimally controlled. There should be mention that the new LNAA may improve adherence as the authors highlight this throughout the abstract and methods.

Methods

Section 2.1.

There seems no real inclusion criteria outside of the fact that some patients were very poorly controlled. Please provide more details about the inclusion and exclusion criterion. Why were 57 patients recruited, but only 12 studied? Is the first paragraph needed – it does not add a great deal where the authors could begin section 2.1 by commenting that 12 sub-optimally controlled patients were recruited. To give the paper further context please define what poor adherence is. Was no dietary information collected whatsoever? This is likely the papers biggest limitation. Despite good adherence to the new LNAA supplement, patients’ habitual diet will play a huge role in patient’s blood results. It is extremely difficult to establish whether the results of this study are a cause of supplementation or diet. Although the trial was long in duration, even monthly completion of 24hr dietary recall would help readers further understand the results. Was the trial registered on a publicly accessible database (i.e. clinicaltrials.gov)? If so, please include the relevant trial registration number in the ethics part of this paragraph.

Section 2.2

Line 95. Who prescribed the product? Please present the nutritional information of the product in the manuscript. This is important information that may be missed when kept as supplementary. Line 99-110. Wording is informal, please amend.

Section 2.3

Ensure further detail about the intervention period is added to the abstract. What was the intra-assay coefficient of variation of Phe and Tyr levels? This should be added to show the accuracy of the blood results. Line 111. Remove the word equipment

Section 2.4

If fortnightly measures of Phe and Tyr were taken throughout the 12-month period, why do authors only explore disclose baseline and after the 12-month period? What happened during the 12 months? Including the fortnightly measures, while needing differing statistical analysis, would surely strengthen the study. Is the t-test performed on baseline and mean intervention values? Or baseline vs month 12? This is not clear in this section and makes the interpretation of results difficult. Please make clearer throughout the manuscript.

Results

Line 135-137. The failure to collect dietary information is one of the main limitations of this study. Weight remains very stable throughout the 12-month period, but as to whether this is due to dietary factors is difficult to establish. Line 141.142. “The frequency of DBS ranged from 14–39 times over the 12-month treatment period.” Can the authors expand further on the above sentence? If patients were compliant with DBS and completing them fortnightly as stipulated in the manuscript, I believe patients should have taken 26 DBS overall. The range presented seems to indicate that some patients may not have been fully compliant with DBS and some over-compliant? The compliance of patients with DBS, perhaps as a percentage, should be included in the text. Table 3. Are the values communicated in table 3 an average of values over the 12-month intervention period? Make this clear. DBS frequency column. It seems a track change has not been accepted/declined. Furthermore, while there is an Asterix, there is no footnote to expand on what it means. Figures 1, 2 and 3. Why do the authors present individual data yet communicate on means. The text surrounding individual responses should be kept, but I recommend the figures are changed to show group mean at baseline and after the 12-month period. The significance for tyrosine will be more visible. Line 167-174. If the authors can graphically depict and communicate on data from baseline vs 6 months and 12 months, why did the authors not perform statistical analysis of these data? Analysis should be performed and included in the paper. There is no mention of compliance to the intervention period. This needs to be included in the paper.

Discussion

The discussion is long. Can authors be more concise with the discussion and continually link back to the findings of this study. At times the discussion seems disjointed. Line 202-203. “This study identified a reduction of plasma Phe levels with LNAA supplementation, and proposed that this was due to competition with Phe at the level of transport across the gut.” The latter part of this sentence needs to be weakened. Without direct measurement of brain/gut levels the findings can only be speculative of this. Line 212. Would the word ‘follow on’ be more appropriate that “following”? Line 247-248. This is the first mention of Phe levels 12 months before the intervention. Why do the authors not provide this data either in the methods or results to provide a 12 month by 12-month comparison? Line 249-250. Although not with LNAA, a recently published study (Green et al 2019; https://doi.org/10.3390/nu11092035) also showed no change on Phe levels with a protein substitute in patients with suboptimal compliance, but improved nutritional intake. Cite this paper to show this issue is not restricted to suboptimal patients only taking LNAA. Line 260-262. Considering the reduction in Phe from baseline to 6 months, this provides greater justification to include in this data throughout the paper. It would be interesting to learn why Phe decreased and then returned to baseline; dietary reasons? compliance issues? Line 273 -275. If the LNAA is high in Tyr perhaps the significant increase over the 12 months is unsurprising. However, the strength of this statement is again unknown due to the lack of dietary intake and compliance data reported. This paper discloses no limitations and there are many. Please add the limitations of this to the discussion. Likely the biggest limitation of this paper did not make any effort to measure dietary intakes. Many of the results are meaningless without this information. Dietary intakes may be the biggest influencing factor of the results, yet without communication of this data it remains unknown. It is a very small cohort and there is no control group.

Conclusions

Line 291-292. No data concerning compliance for DBS of taking LNAA was provided and therefore the authors cannot justify this conclusion.

Round 2

Reviewer 4 Report

Thank you for the opportunity of review the revised paper of Burlina et al. The revised manuscript much improved and I thank authors for taking my feedback into account when revising. My concerns have been met adequately and I am now happy to accept this paper for publication.